# Revisiting John Snow to Meet the Challenge of Nontuberculous Mycobacterial Lung Disease

**DOI:** 10.3390/ijerph16214250

**Published:** 2019-11-01

**Authors:** Aashka Parikh, Christopher Vinnard, Nicole Fahrenfeld, Amy L. Davidow, Amee Patrawalla, Alfred Lardizabal, Andrew Gow, Reynold Panettieri, Maria Gennaro

**Affiliations:** 1New Jersey Medical School, Rutgers, The State University of New Jersey, Newark, NJ 07103, USA; app101@njms.rutgers.edu (A.P.); patrawam@njms.rutgers.edu (A.P.); lardizaa@njms.rutgers.edu (A.L.); 2Public Health Research Institute, Rutgers, The State University of New Jersey, Newark, NJ 07103, USA; gennarma@njms.rutgers.edu; 3School of Engineering, Rutgers, The State University of New Jersey, Piscataway, NJ 08854, USA; nfahrenf@rutgers.edu; 4School of Public Health, Rutgers, The State University of New Jersey, New Brunswick, NJ 08854, USA; davidoal@sph.rutgers.edu; 5Ernest Mario School of Pharmacy, Rutgers, The State University of New Jersey, Piscataway, NJ 08854, USA; gow@pharmacy.rutgers.edu; 6Rutgers Institute for Translational Medicine and Science, Rutgers, The State University of New Jersey, New Brunswick, NJ 08854, USA; rp856@rbhs.rutgers.edu

**Keywords:** mycobacterial infections, surface water, opportunistic plumbing pathogens, geographic distribution

## Abstract

Nontuberculous mycobacteria (NTM) are ubiquitous components of the soil and surface water microbiome. Disparities by sex, age, and geography demonstrate that both host and environmental factors are key determinants of NTM disease in populations, which predominates in the form of chronic pulmonary disease. As the incidence of NTM pulmonary disease rises across the United States, it becomes increasingly evident that addressing this emerging human health issue requires a bold, multi-disciplinary research framework that incorporates host risk factors for NTM pulmonary disease alongside the determinants of NTM residence in the environment. Such a framework should include the assessment of environmental characteristics promoting NTM growth in soil and surface water, detailed evaluations of water distribution systems, direct sampling of water sources for NTM contamination and species diversity, and studies of host and bacterial factors involved in NTM pathogenesis. This comprehensive approach can identify intervention points to interrupt the transmission of pathogenic NTM species from the environment to the susceptible host and to reduce NTM pulmonary disease incidence.

## 1. Introduction

The prevalence of pulmonary infections caused by nontuberculous mycobacteria (NTM) is increasing [1,2,3,4,5,6,7]. The burden of NTM infections on the U.S. healthcare systems was estimated to cost $815 million annually [8]. Many NTM species, including the most common pathogens, are ubiquitous in both surface water (rivers, lakes) and water distribution systems, sharing the designation of “opportunistic premise plumbing pathogens” with bacteria such *Legionella pneumophila* and *Pseudomonas aeruginosa* [9].

Epidemiologic and environmental studies of NTM pulmonary disease often travel in non-intersecting paths. Observational clinical studies provide insight into baseline patient characteristics associated with the development of NTM pulmonary disease or the likelihood of a favorable response to NTM-directed treatment. Simultaneously, environmental sampling has been used to define potential mechanisms of exposure for pathogenic NTM species. Often, a genetic link can be established between the infecting NTM strain isolated from the patient and the NTM strains in the household water source. However, there remain tremendous knowledge gaps regarding the continuum that links the environmental reservoir of NTM species to clinical disease in susceptible hosts. Consequently, basic questions regarding prevention strategies and therapeutic interventions remain unanswered [10,11].

A comprehensive framework to address the rising challenge of NTM pulmonary disease must combine population-based clinical epidemiology with community-based environmental assessments into a single research framework. This approach has its origin in the classic “shoe leather” epidemiology established in the mid-19th century by one of the founders of modern epidemiology, Dr. John Snow, as he pursued the cause of an epidemic of diarrheal disease in London. Snow was skeptical of the “miasma” theory of disease, and suspected that diarrheal disease in the London Soho neighborhood was caused by the intake of contaminated water from the Thames River into the pumping system destined for household use. By creating dot maps of incident cholera cases in the community, he identified a single Broad Street water pump as a point source of the disease, and successfully ended the epidemic simply by removing the pump handle. Although disease mapping tools have become much more sophisticated since the dot maps created by Snow, the objectives of disease mapping remain the same. The identification of geographic clusters of disease frequently provides insight into the underlying mechanisms of disease transmission in the population.

The discipline of spatial epidemiology can be defined as the “description and analysis of geographically indexed health data with respect to demographic, environmental, behavioral, socioeconomic, genetic, and infectious risk factors” [12]. Such an approach is particularly well-suited to address the challenge of NTM pulmonary disease. Our understanding of NTM biology and its environmental niche, combined with increasing evidence of spatial clusters of NTM pulmonary disease across the United States, demonstrate the need for a research framework that incorporates the host risk factors for NTM pulmonary disease alongside the determinants of NTM residence in the environment. The spatial variation of NTM pulmonary disease will be related both to the environmental-level variability of soil and water characteristics that support NTM growth in the environment, as well as engineering-driven aspects of water distribution systems that preferentially select and promote NTM species at the human–microbe interface. 

Furthermore, clustering of NTM infections, whether in time, space, or among certain individuals in a population, may identify additional causal factors that suggest interventions to be applied at different levels. A comprehensive population-based approach, combining environmental and clinical data, should include standardized assessments of patient-associated factors related to NTM pulmonary disease acquisition and clinical response, characterization of environmental distribution of pathogenic NTM species in water distribution systems and household water supplies, and ascertainment of specific environmental characteristics that support NTM growth and propagation in the underlying soil and surface water. 

In this review, we will follow the path traveled by pathogenic NTM species from the environmental reservoir to the patient, identify aspects of NTM microbiology and clinical epidemiology essential to a population-based NTM research program, and review recent efforts to understand the spatial epidemiology of NTM pulmonary disease based on these characteristics. With the integration of environmental and host factors, interventions designed to limit the spread of NTM can then be planned and evaluated. 

## 2. A Microbial Niche in Soil and Draining Surface Waters

Contamination from soil to surrounding water bodies establishes the primary environmental reservoir for transmission of pathogenic NTM species to humans [13,14,15,16,17,18,19]. First, properties of soil are major determinants of NTM prevalence in microbial communities. In soil, the content of sodium, copper, and silt promotes growth of NTM species, while manganese and clay content inhibits growth [20,21]. High atmospheric water content may also promote NTM growth in soil [16]. In this manner, geographic variability in soil composition imposes an initial determinant of NTM prevalence in the environment (Figure 1). 

Once established in soil, additional environmental factors support the growth and propagation of NTM in the draining surface waters. Drivers of NTM growth in surface waters include low pH [13]; water temperature up to 55 °C [22]; low dissolved oxygen content [14]; and a high content of salt [23], soluble zinc, humic acid, and fulvic acid [24]. These atmospheric and surface water properties introduce a second layer of geographic variability into the factors that support NTM growth in the environment, such as evapotranspiration, the process by which water is transferred from the land to the atmosphere by evaporation from the soil and other surfaces and by transpiration from plants (Figure 2). 

As a consequence of lipid-rich outer membranes composed of mycolic acids [19], NTM species are hydrophobic [25], impermeable [26,27], and relatively slow-growing [26]. These biological properties allow NTM species to attach to surfaces and form biofilms, supporting adherence to rocks, plant material, and other environmental substrates in bodies of surface water [28,29]. The combination of a slow growth rate and cellular impermeability is advantageous under conditions of nutrient starvation or toxin exposure [30]. Moreover, NTM species are capable of intracellular growth in water-associated protozoa and amoebae, which can promote virulence and perhaps ‘train’ NTM species to grow in animal macrophages [17,18].

By adhering to surfaces, remaining impermeable to toxins, growing intracellularly, surviving under stress conditions, and maintaining comparatively slow growth rates overall, NTM species are ideally suited to thrive in surface water environments. Notably, surface water sources, as opposed to groundwater from wells, provide the majority of public-supply freshwater in the United States [31]. However, NTM species have been reported in public [32,33] and private distribution systems [32,33] transmitting treated groundwater [34], as well as private wells [33,35]. NTM species were observed in groundwater-fed drinking water systems at comparable rates to those fed by surface water [36]. It is possible that similar selecting conditions present in treated groundwater systems as treated surface water systems (i.e., disinfection removing competition, long residence time) are providing suitable conditions for NTM proliferation despite differences in source water microbial ecology. Households with water from treated systems, whether public or private, had a higher relative risk of NTM observation in water than in water from private wells [32]. Likewise, *Mycobacterium avium, M. intracellulare,* and *M. scrofulaceum* were rarely detected in a survey of untreated southern U.S. groundwater [37].

## 3. The Role of Household Water Systems

Understanding how NTM pulmonary disease has become a public health threat requires a detailed understanding of how these bacteria have progressively gained a foothold in water systems of the United States and elsewhere. Intake pipes draw surface water into treatment facilities, followed by distribution into municipalities and eventually households, through intake pipes, and ultimately in household faucets and showerheads [38,39]. NTM species persist in drinking water sampled from point-of-use sites at cold water outlets despite the introduction of ozonation and filtration systems into the water treatment facility [40,41]. Cellular impermeability confers resistance to commonly used water disinfectants such as chlorine [42,43,44,45,46], chlorine dioxide [45,46], and chloramine [45,46]. Some NTM species are 100-fold more resistant to these disinfectants than *P. aeruginosa* and *E. coli*, and thus disinfectant use in water systems can select for NTM species over chlorine-sensitive competitors [46]. 

The ability to form biofilms provides NTM species with a competitive advantage over other microorganisms in adhering to surfaces within pipelines [47,48] and household plumbing systems [49]. As a result, biofilm-associated NTM microbial communities can become long-term, stable components of water distribution pipelines, shedding bacteria downstream to the human–microbe interface [41,50]. NTM species have been isolated from biofilms within plumbing systems composed of stainless steel, glass, zinc-galvanized steel, and polyvinyl chloride (PVC), and are particularly well-adapted to growth on copper surfaces [51]. As evidence of this selection process, the NTM species comprise a greater component of household water microbiome as compared with their proportion in the source surface water [52]. This selection process is further amplified by longer durations of water stagnation time [52]. 

A final NTM enrichment step is driven by the household water heater. In many ways, household water heater systems provide an ideal habitat for heat-tolerant NTM species: warm water, suitable surface areas for biofilm formation, and little competition [53]. For many plumbing pathogens, such as *Legionella pneumophila*, exposure to water temperatures greater than 46.1 °C (115 °F) leads to decreased survival, and above 50.0 °C (122 °F) leads to cell death [54]. In contrast, some NTM species can tolerate the set-point temperatures found in many household water heaters [55]. In one study, household water heaters maintained at temperatures less than 52.2°C (126 °F) were more likely to yield NTM species compared with households that maintained water heater temperatures greater than 54.4 °C (130 °F) [32]. Thus, water heater temperatures in a certain range provide selective pressure that disproportionately promotes NTM growth [48,56,57]. These are critical observations in light of the ongoing debate regarding the most appropriate set-point for the household water heater for the prevention of scalding injuries [58]. For example, the U.S. Consumer Product Safety Commission recommends a manufacturer reduction in temperature set-point of 48.9 °C (120 °F) [59].

## 4. Transmission from the Microbial Reservoir to the Individual

With repeated exposure to NTM-contaminated water in the household, infection may be established through two potential mechanisms: inhalation of aerosolized droplets containing NTM or ingestion with subsequent aspiration. NTM species easily aerosolize from water thanks to their hydrophobic cellular surfaces [60,61], and can be detected in water aerosolized from showerheads [28], taps [28], humidifiers [28], and heating/ventilation systems [28]. The concentration of NTM in aerosolized water has been shown to be 1000 to 10,000 times higher than in the cellular suspensions from which they arise [60]. They are found in aerosolized water droplets small enough to enter the alveoli, likely a dominant mechanism that drives pulmonary disease acquisition [60]. Ingestion and chronic aspiration is the more likely mechanism among individuals with gastric reflux disease [62,63]. 

Using environmental sampling in the household water supplies of patients with established NTM disease, DNA sequencing has confirmed that the NTM patient isolate matches the household isolate in many instances. Bacterial isolates from drinking water as well as hot tubs [64], bathroom inlets [65,66], showerheads [67], and spa pools [68] from facilities [69,70,71] or homes [64,65,66,67,69] of patients with an NTM infection have been shown to match the NTM isolates from the patient [72]. It must be noted that genotyping has sometimes found related, but non-identical matches between the environmental isolates and clinical isolates, which may be a consequence of the high degree of genetic variability in NTM colonies [67,73,74].

Recently, direct patient-to-patient transmission of highly pathogenic NTM species has also been described in the cystic fibrosis patient population. Direct transmission of *M. abscessus* was identified by detection of nearly identical strains in a cohort of cystic fibrosis patients at a single clinical site [75]. Subsequently, genotyping of clinical isolates of *M. abscessus* from cystic fibrosis centers worldwide identified three *M. abscessus* clones responsible for the majority of pulmonary disease. Furthermore, these clustered *M. abscessus* isolates demonstrated increased virulence compared with unclustered isolates [76]. Whether these observations represent a phenomenon unique to the cystic fibrosis patient population, rather than a more general mechanism of virulent *M. abscessus* disease transmission, remains to be determined.

Hospital-related infections [77,78,79] and outbreaks [80,81], owing to various forms of contaminated water [81,82,83,84] or improper disinfections [85], have been reported. Pathogenic NTM species have been cultured from hospital water reservoirs, where their heat resistance allows them to survive in temperatures up to 55 °C [57,81,82,83]. NTM species that are not heat-tolerant can colonize cold water distribution systems in healthcare facilities [81,86,87,88,89,90,91]. As NTM can also grow in distilled water, presumably using carbon compounds leaching out from the water container, contaminated distilled water sources have also been implicated in nosocomial NTM disease outbreaks [92,93,94]. Not surprisingly, there have been numerous reports of nosocomial NTM disease outbreaks favored by the intrinsic properties of these bacteria, including pulmonary NTM disease presentations [95,96]. The additional contribution of antibiotic overuse as a selective pressure on NTM strains in nosocomial environments, including drug-resistant NTM strains, has not yet been unexplored. More work is needed to link NTM strains in the nosocomial environment and the strains found in community water distribution systems.

## 5. At-Risk Individuals with Repeated NTM Household Exposures: A Perfect Storm

Clinical experience demonstrates that NTM exposure is necessary, but not sufficient to establish NTM infection in the human host. Widespread exposure to NTM and periods of intermittent colonization are suggested by the presence of antibodies against NTM in various populations. Although NTM exposure may be widespread in certain geographic clusters, owing to numerous mechanisms discussed above, progression from NTM exposure to disease is related in part to underlying co-morbidities or immunologic deficiencies. Many of these chronic underlying diseases that promote NTM disease also demonstrate geographic variability across the United States (such as chronic obstructive pulmonary disease (COPD)), which adds an additional layer to the uneven geographic distribution of NTM disease in NTM-exposed populations (Figure 3).

Primary immunologic and pulmonary diseases associated with NTM disease acquisition have been extensively reviewed elsewhere [97]. Higher rates of progression to NTM infection in individuals with defects in immune response have identified essential pathways, signaling molecules, and effectors of NTM disease protection [95], including interleukin (IL)-12 [98,99,100], interferon-gamma (IFNγ) [101,102,103], STAT1 gene [104,105,106,107,108], TYK2 gene [109], IRF-8 gene [110], ISG-15 gene [111], and RORC gene [112]. Underlying lung conditions, particularly cystic fibrosis [113,114,115], COPD [116,117,118], α-1-antitrypsin (AAT) anomalies [119], chronic bronchiectasis [120], pneumoconiosis [121,122], primary ciliary dyskinesia [123], calcified chest adenopathy [124], and pulmonary alveolar proteinosis [125,126,127,128,129,130,131,132], have all been identified as risk factors for acquisition of NTM disease. Systemic illnesses that target the immunologic pathways involved in response to NTM infection [2,30], or therapies that inhibit individual components of these pathways, have also been implicated as risk factors for NTM disease progression [97]. These include the receipt of tumor necrosis factor alpha (TNFα)-antagonist therapies [133,134], immunosuppressive regimens [135], B cells lymphocyte suppressors [136], and immunosuppressive therapies in the setting of solid organ transplantation [137]. Increasing the use of these particular classes of immunosuppressant therapies may contribute to the rise in the prevalence of NTM lung disease [138].

Notably, many cases of NTM pulmonary disease occur in individuals without defined risk factors [97]. Thin, postmenopausal women, with either white or Asian/Pacific Islander race/ethnicity, have increased risk of pulmonary NTM disease for unknown reasons. Estrogen has been shown to bind macrophages and augment their phagocytic function, and the reduced estrogen levels in postmenopausal women have been speculated as an underlying cause of increased rates of pulmonary infection in postmenopausal women [139,140]. The mechanisms that promote infection among exposed individuals without identifiable risk factors, and likewise the subsequent progression of disease, are likely to be multifactorial. Both host-related and pathogen-related factors likely determine whether environmentally driven repeated exposures eventually lead to an established infection [141].

## 6. Evidence of Geographic Clustering of NTM Disease

It is clear that the distribution of NTM disease across the United States follows a non-uniform pattern, with several geographical clusters clearly identified. In a population-based analysis of Medicare data, with NTM disease identified by a single diagnosis code, geographic clusters of NTM disease were identified in California, Florida, Hawaii, Louisiana, New York, Oklahoma, Pennsylvania, and Wisconsin (Figure 4) [6]. These clusters had a tendency to include urban areas with comparatively higher education and income levels, although this distribution may also be influenced by underlying patterns of Medicare eligibility in the U.S. population. From an environmental perspective, cluster regions had higher mean daily evapotranspiration levels and higher percentages of area covered by surface water. Soil content in these areas includes greater copper [20] and sodium levels [20] and lower manganese levels [20]. As discussed above, these are environmental conditions that appear to promote NTM growth and persistence in surface water bodies. 

Our understanding of the spatial epidemiology of NTM disease is beginning to achieve more granular levels of analysis with regards to environmental determinants. In a Colorado-based study, NTM infections were distributed in clusters that shared a common water source [21]. In addition to the role of air and water temperatures, this approach confirmed that soil acidity and silt content were significantly associated with disease risk, while manganese and clay content were protective against NTM disease [21]. Although water has been uniformly implicated in harboring NTM, these findings illustrate that soil characteristics are the underlying determinant of NTM growth in the environment, with downstream distribution from soil into the draining surface waters. Similar relationships between soil composition and NTM disease clusters were observed in Queensland, Australia [142].

Mapping water distribution systems in conjunction with environmental and clinical data will provide additional insights into NTM disease mechanisms. Environmental sampling studies have demonstrated that the abundance of NTM species in water systems increases in proportion to the distance from the point of entry into the water distribution system, which reflects the residence time in the system [41,52]. As distance increases, the genetic diversity of NTM species decreases overall, with increasing abundance of pathogenic NTM species, including *M. avium* [52]. This may be a consequence of increased stagnation time in the water pipes and proliferation of slow-growing *M. avium* in biofilms, as discussed above. Similarly, the Colorado-based geographic study observed increasing rates of NTM infection with increasing distance from the treatment plant to the household [21,52]. These are intriguing initial observations of the potential role of the water distribution system geography (including source water type, service areas, and stagnation times) in driving pathogenic NTM species prevalence in household water supplies, leading to colonization at the human–microbe interface in faucets and showerheads. 

## 7. Summary

Studies in the field of NTM biology and disease have predominantly focused on either the patient characteristics that promote the acquisition of NTM infection and disease progression, or the environmental drivers of NTM growth and propagation in soil and water systems. We propose that addressing the emerging public health threat of NTM pulmonary disease depends upon a multi-disciplinary research framework that includes the assessment of environmental characteristics promoting NTM growth in soil and surface water, evaluation of associated water distribution systems (piping, chlorination, geography), and direct sampling of water supplies for NTM contamination and diversity. These environmental-based studies should complement investigations of the clinical epidemiology of NTM pulmonary disease within populations drawn from the same geographic areas, and may shed light on poorly understood phenomena such as relapse or re-infection following treatment. For example, prospective cohort studies of patients with NTM pulmonary disease could routinely include environmental sampling in the patient’s household, performed at regular intervals during follow-up, in order to disentangle patient-focused and environment-focused drivers of treatment response or disease relapse. This unified framework will be required to identify the NTM “pump handles” for removal and interrupt the continuum that leads from NTM soil colonization to clinical disease. A comprehensive approach that follows the lessons of John Snow is essential to reverse the trends of NTM pulmonary disease in the 21st century and beyond.

## Figures and Tables

**Figure 1 ijerph-16-04250-f001:**
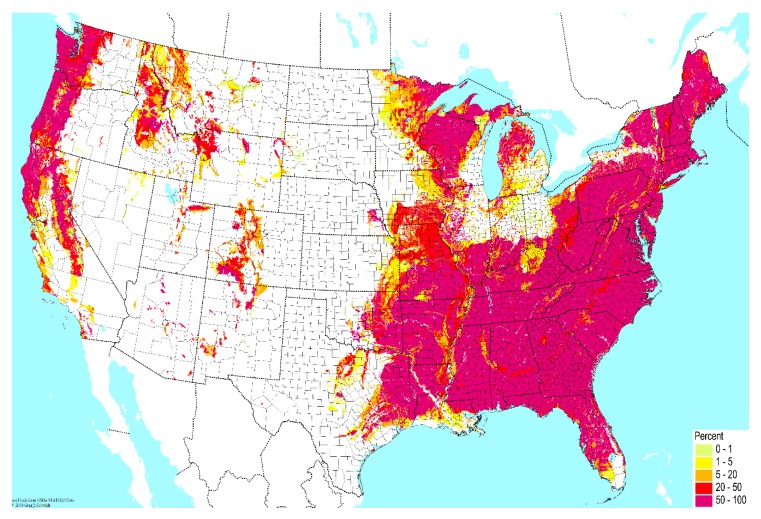
Geographic variability across the United States in the percent of very poorly-drained soil. Data from the United States Department of Agriculture USDA), National Resources Conservation Service (NCRS); reproduced with permission from the Biota of North America Program.

**Figure 2 ijerph-16-04250-f002:**
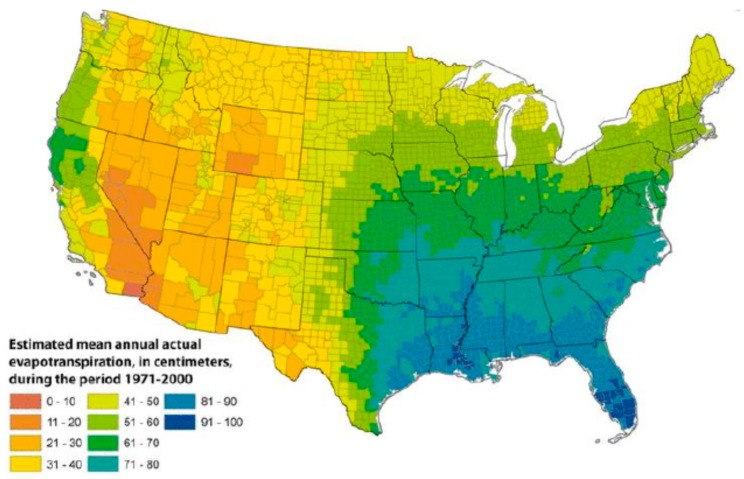
Geographic variability in evapotranspiration levels in the United States, 1971–2000; reproduced with permission from Sandford et al., 2012.

**Figure 3 ijerph-16-04250-f003:**
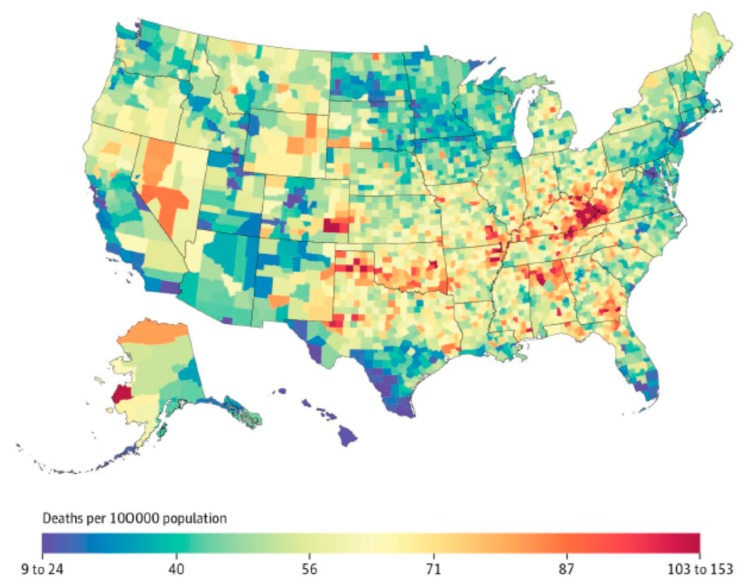
Age-adjusted standardized mortality rate from chronic obstructive pulmonary disease in the United States, 2014; reproduced with permission from Dwyer-Lindgren et al., 2017.

**Figure 4 ijerph-16-04250-f004:**
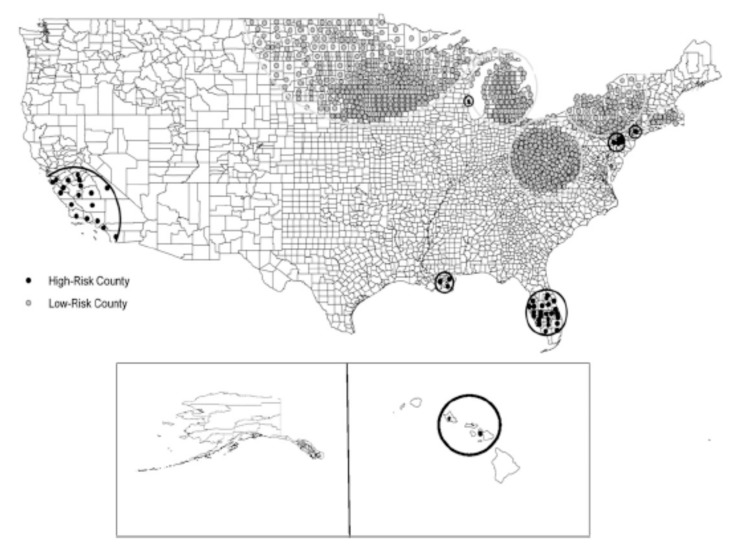
Significant clusters of counties identified as being at high or low risk for pulmonary NTM disease among U.S. Medicare beneficiaries 65 years of age and older; reproduced with permission from Adjemian et al., 2012.

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
