# Peer review of "Revisiting John Snow to Meet the Challenge of Nontuberculous Mycobacterial Lung Disease"

_ijerph, 2019, doi:10.3390/ijerph16214250_

Round 1

Reviewer 1 Report

Authors reviewed the environmental drivers and host factors of pathogenic NTM growth and propagation, proposed multi-disciplinary interventions/strategies to reverse NTM pulmonary diseases. Generally this is a well-organized and written review elaborating recent efforts to understand the spatial epidemiology of NTM pulmonary disease. The critical analysis of numerous references also testifies the authors’ knowledge and innovation in the field.

The minor concern is whether primary immunodeficiency syndromes or antibiotic abuse may be also important host/environmental factors? Authors may address and discuss this briefly.

Author Response

We thank the reviewer for their time and effort in reviewing our manuscript. We are confident that the manuscript has been improved as a result of this effort.

-Discussion of the effect of primary immunodeficiencies is included in Lines 218-226.

-The reviewer raises an excellent question regarding antibiotic overuse, and the contribution to NTM disease. After an additional literature review, we could identify no data or studies to address this relationship, although we agree it is plausible (similar to the effects of chlorine or chloramine in selecting for NTM in water distribution systems). We have revised the manuscript to include mention of this key knowledge gap (Lines 201-2014).

Reviewer 2 Report

Dear authors,

Thank you for submitting the manuscript.

After careful reading, the manuscript is interesting that focuses on the NTM infection in humans and how the transmission plays its role. There are some comments (in the attached file) provided for further improvement of this manuscript.

Sincerely,

Author Response

We thank the reviewer for their time and effort in reviewing our manuscript. We are confident that the manuscript has been improved as a result of this effort. We have revised accordingly each of the highlighted sections in the reviewer's file.

Reviewer 3 Report

This is a well written and comprehensive review of nontuberculous mycobacterial (NTM) distributions in the USA, aiming to demonstrate the contribution of geographical distribution, environmental factors and host to pulmonary infections caused by NTM. Minor comments are below:

Lines 97-99: this sentence should be expanded to include the definition of 'evapotranspiration', as follows: "These atmospheric and surface water properties introduce a second layer of geographic variability into the factors that support NTM growth in the environment, such as evapotranspiration, the process by which water is transferred from the land to the atmosphere by evaporation from the soil and other surfaces and by transpiration from plants (Figure 2)." Section 6, evidence of geographic clustering, lines 236-245. This data is possibly biased by the quality of Medicare data, which is restricted to those who can afford Medicare in general. Therefore the fact that the results point to "urban areas with comparatively higher education and income levels" could be affected by this bias and this should be mentioned as a limitation in the text. I would have liked to see a section on proposed solutions on how to tackle the problem so comprehensively introduced and described by the authors. However, if this is not possible, a sentence should be included to state this as a limitation.

Author Response

We thank the reviewer for time and effort in reviewing our work. We have revised the manuscript accordingly and are confident the manuscript has been improved.

-We have added a definition of evapotranspiration, as recommended (Lines 98-101).

-We have added a sentence to the summary with a more concrete example of one proposed solution (Lines 293-296).